# A Portrait of Intratumoral Genomic and Transcriptomic Heterogeneity at Single-Cell Level in Colorectal Cancer

**DOI:** 10.3390/medicina57111257

**Published:** 2021-11-17

**Authors:** Andrea Angius, Antonio Mario Scanu, Caterina Arru, Maria Rosaria Muroni, Ciriaco Carru, Alberto Porcu, Paolo Cossu-Rocca, Maria Rosaria De Miglio

**Affiliations:** 1Istituto di Ricerca Genetica e Biomedica (IRGB), CNR, Cittadella Universitaria di Cagliari, 09042 Monserrato, Italy; 2Department of Medical, Surgical and Experimental Sciences, University of Sassari, Via P. Manzella, 4, 07100 Sassari, Italy; scanu@uniss.it (A.M.S.); mrmuroni@uniss.it (M.R.M.); alberto@uniss.it (A.P.); rocco@uniss.it (P.C.-R.); 3Department of Biomedical Sciences, University of Sassari, 07100 Sassari, Italy; 30039590@studenti.uniss.it (C.A.); carru@uniss.it (C.C.)

**Keywords:** colorectal carcinoma, intratumor heterogeneity, single-cell next-generation sequencing, precision medicine

## Abstract

In the study of cancer, omics technologies are supporting the transition from traditional clinical approaches to precision medicine. Intra-tumoral heterogeneity (ITH) is detectable within a single tumor in which cancer cell subpopulations with different genome features coexist in a patient in different tumor areas or may evolve/differ over time. Colorectal carcinoma (CRC) is characterized by heterogeneous features involving genomic, epigenomic, and transcriptomic alterations. The study of ITH is a promising new frontier to lay the foundation towards successful CRC diagnosis and treatment. Genome and transcriptome sequencing together with editing technologies are revolutionizing biomedical research, representing the most promising tools for overcoming unmet clinical and research challenges. Rapid advances in both bulk and single-cell next-generation sequencing (NGS) are identifying primary and metastatic intratumoral genomic and transcriptional heterogeneity. They provide critical insight in the origin and spatiotemporal evolution of genomic clones responsible for early and late therapeutic resistance and relapse. Single-cell technologies can be used to define subpopulations within a known cell type by searching for differential gene expression within the cell population of interest and/or effectively isolating signal from rare cell populations that would not be detectable by other methods. Each single-cell sequencing analysis is driven by clustering of cells based on their differentially expressed genes. Genes that drive clustering can be used as unique markers for a specific cell population. In this review we analyzed, starting from published data, the possible achievement of a transition from clinical CRC research to precision medicine with an emphasis on new single-cell based techniques; at the same time, we focused on all approaches and issues related to this promising technology. This transition might enable noninvasive screening for early diagnosis, individualized prediction of therapeutic response, and discovery of additional novel drug targets.

## 1. Introduction

Genome and transcriptome sequencing and editing technologies, supplemented with machine learning, are setting the stage for the transition from traditional to precision medicine [1,2,3,4,5,6,7,8]. In cancer studies, we are observing a promising transition from research on spatiotemporal tumor heterogeneity [9,10,11] to early-stage clinical trials [12,13,14,15,16,17].

As inter-tumor heterogeneity is characterized by variability in patients with the same histologic type [18,19], this might influence clinical care in cancer by providing targeted therapies based on tumor genetic features. We can now monitor clonal dynamics during treatment or identify clinical resistance during disease progression.

Intra-tumoral heterogeneity (ITH) is detectable: subpopulations of cancer cells differ in genome features and tumor areas and/or may evolve/differentiate over time [20,21,22]. Thus, ITH represents a key determinant of treatment failure, drug resistance, and disease recurrence [19].

Colorectal carcinoma (CRC) is a leading mortality cause worldwide [10,11] and is characterized by heterogeneous genomic, epigenomic and transcriptomic alterations [23,24,25,26,27,28,29]. The heterogeneous nature of CRC may also be related to colorectal cancer stem cells (CCSCs): a small population with stem-like behavior responsible for tumor progression, recurrence, and resistance to therapy [16].

CRC treatment has been standardized based on clinicopathological and genetic features (KRAS/NRAS/BRAF mutation and Microsatellite instability (MSI) status), as well as based on tumor staging. Characterization of multiple samples from the same patient proved to be a significant ITH indicator between different areas of the same tumor (spatial heterogeneity) as well as comparing the primary tumor and a subsequent local or distant recurrence (temporal heterogeneity) [18].

The ability of next-generation sequencing (NGS) both at whole genome and single-cell levels to identify disease-associated variants and tumor features triggered a renewed interest on the effectiveness of biomedical and oncology research [1,2,18,30]. Whole genomic and transcriptomic profiling only shows us the average cellular characteristics, thus hiding critical aspects of tumor heterogeneity. Deep bulk sequencing can only capture 1% of the cell population, excluding some types such as circulating tumor cells. Therefore, single-cell techniques allow us to accurately explore cellular properties [31]. Despite recent advances, single-cell next generation sequencing (scNGS) suffers from limited availability of public data/databases and the lack of standardization of laboratory protocols and computer analysis.

Although over the years conventional research has improved, as well as outcomes in CRC patients through diagnosis standardization, staging, and multimodal treatment, important critical and clinical issues remain unresolved [6,7,8,9,32].

Recent considerations of dynamic clonal evolution [33], spatiotemporal detection of genomic clones, circulating tumor DNA (ctDNA), identification of ITH [34] and circulating cell heterogeneity [35] allow delineation and improvement of therapeutic failure and relapse [36]. Single-cell transcriptomics, CRISPR-Cas9, and their combination returned exciting data on cell-to-cell drug-dependent variability [9,37,38]. Pioneering combinations of scNGS, CRISPR-Cas, and Hi-C technologies raise high hopes for understanding the linear and nonlinear interactions that control gene expression at single-cell resolution [39].

Based on a review of published data, we aimed at discussing the possible achievement of a transition from CRC clinical research to precision medicine with a special emphasis on new single-cell-based techniques, focusing on all approaches and issues related to these technologies. This transition may provide feasible non-invasive screening procedures for early diagnosis, individualized prediction of therapeutic response and discovery of additional novel drug targets.

## 2. Innovative Methodologies Applied to Precision Medicine

Proper analysis and extensive use of the large amount of data generated from single scNGS experiments are very challenging and require experienced personnel. A full understanding of the experimental and computational pathways starting from the wet lab to the sophisticated computer analysis of data is needed. Attention must be given to quality control measures for determining which individual cells to include for further examination, data normalization methods, clustering, and visualization for dimensional reduction of data into a two-dimensional graph.

As far as the experimental design is involved, no less significant are the costs that vary from EUR 1–2 to a few cents per cell. The price is highly dependent on the number of cells sequenced, the desired sequencing depth, and the sequencing platform used.

Regardless of cell separation method and labeling of mRNA molecules, all approaches rely on similar computational pipelines for transcriptional profiling. Some concepts are applicable to the majority of single-cell sequencing platforms that use DNA barcodes as an approach to link mRNA transcripts to a single-cell source. Single-cell technologies can be used to define subpopulations within a known cell type by seeking differential gene expression within the cell population of interest; at the same time, they can effectively isolate signal from rare cell populations that would not be detectable by other methods. Each individual cell analysis is driven by clustering of cells based on their differentially expressed genes. The genes driving the clustering can be utilized as unique markers for a specific cell population.

### 2.1. Generation of Single-Cell Expression Datasets

There are several high-throughput single-cell sequencing platforms on the market at the moment: the most widely used and cost-effective are Fluidigm C1, DropSeq and Chromium 10X [40,41]. These technologies can define the transcriptional profile from hundreds to thousands of individual cells simultaneously. They all are based on labeling mRNA molecules with DNA barcodes during reverse transcription and/or subsequent steps, which allow for indexing of transcripts to their individual cells of origin [42,43,44].

The various cell capture methods have to consider several parameters that differ from one method to the other and affect the final sequencing results. The main parameters are the number of starting cells (which varies from about 1000 to 500,000), the method of cell separation (cell capture, droplet-based, etc.) and the efficiency of cell capture [45].

The C1 system isolates single cells into individual reaction chambers in the Fluidigm integrated fluidic circuit (IFC). The optically clear IFC enables the operator to automatically stain captured cells and examine them by microscopy for viability, surface markers or reporter genes. Cell lysing, reverse transcription, and cDNA amplification are performed on the C1 Single-Cell Auto Prep IFC using a SMARTer Ultra Low RNA Kit for cDNA synthesis [46,47,48,49] followed by a standard Illumina NGS library protocol.

Droplet-based single-cell gene expression approaches, including DropSeq and the 10X platform, use microfluidic chips to isolate single cells along with individual microspheres embedded in oil droplets using a microfluidic so that each droplet contains a single cell [50,51]. The microspheres are coated with DNA oligos that are composed of a poly(T) tail at the 3′ end for capturing cellular mRNA, and at the 5′ end possess a cellular barcode that is identical for each oligo coating a single bead and an individual unique molecular identifier (UMI) barcode for high diversity [52,53,54]. The transcripts from each individual cell captured and labeled by the DNA oligos attached to a bead are reverse transcribed and amplified with PCR; subsequently, they are sequenced using a high-throughput platform after breaking and pooling droplet contents.

### 2.2. Bioinformatics Approaches to Single-Cell Analysis

scRNA-seq data analysis poses several unique computational challenges that need the adaptation of existing workflows, as well as the development and application of new analytical strategies (Figure 1). Many analytical procedures rely on specialized algorithms developed and made available to the international community by reference bioinformatics laboratories [55]. Sequencing data from various methods are mostly produced using standard NGS methodology and Illumina instrumentation. They are aligned to a reference genome to annotate each transcript to its gene. Cell barcodes allow computational linkage of each gene transcript to its cell of origin. The number of individual gene transcripts expressed in each cell is counted using UMIs, allowing the assembly of digital gene expression arrays (DGEs), which are tables of cell barcodes and gene counts.

Single-cell experiments can be considered as thousands of separate experiments, so it is essential to apply the right quality control (QC) metrics to decide which individual data sets are valid [56]. For example, in a droplet-based experiment the QC can effectively determine, by applying a number of different parameters, which droplets are failed and exclude these data from further analysis [57,58]. An important QC metric to evaluate is the number of transcripts per cell, or the percentage of transcripts per cell that align with the reference genome and establish a cutoff to identify outliers. These cutoffs must be defined by the user for each experiment, for instance: cells with a few dozen transcripts and/or with several thousand; otherwise, they can be automatically defined by a software as cells with a summary of transcripts greater than two SD from the mean value. Excessive numbers of uniquely barcoded transcripts may result from duplicates (i.e., two or more cells suspended in a drop), whereas a small number of transcripts is an indicator of poor capture quality. Additional QC metrics related to the diversity of the tissue to be analyzed must then be applied [59,60]. For example, in an experiment to study circulating tumor cells, the number of tumor cells will be very low compared with normal blood cells and transcript counts will need to be adjusted; in fact, normal and generally quiescent blood cells have relatively low amounts of RNA compared with active tumor cells.

A common QC metric is the number of mitochondrial gene transcripts: excessive numbers of mitochondrial transcripts indicate cellular stress. In normal tissues, cells with excessive mitochondrial gene expression are not included in the analysis [61]. However, this parameter is highly dependent on the tissue and the purpose of the investigation [62]. Mitochondrial mRNA percentages should be assessed in a tissue-dependent approach.

An important point in analyzing single-cell data is normalization to eliminate batch effects if multiple sequencing runs are to be compared. These batch effects can be caused by a non-avoidable number of technical variations given by different experimental sessions (e.g., RNA isolation method, sequencing depth, etc.). In addition, for bulk RNA sequencing, data normalization involves comparing multiple batches of biological material; however, in sequencing individual cells that are not all the same type, normalization parameters are required to maintain cell-to-cell variability. A common way to normalize sequencing data is based on comparison with housekeeping genes [62,63]. Based on the characteristics of the biological sample, a selected housekeeping gene is chosen for normalization. Assuming that this gene is expressed at the same level in all cells, data are scaled to make the expression level of the housekeeping gene equal in all cells.

The next analytic step is to use a clustering algorithm to determine which cells are closely related. The most widely used is the principal component analysis (PCA) [60], which uses a relatively simple linear dimensionality reduction algorithm; the latter can predict the relatedness of cells in this case based solely on differential gene expression. Due to the highly dimensional nature of scRNA-seq data, several reduction methods are required, including nonlinear methods such as the t-Distributed stochastic neighbor embedding (t-SNE) and the uniform manifold approximation and projection (UMAP) techniques. The t-SNE is a common data visualization approach [64,65,66] that uses a machine learning algorithm to reduce size and is suitable for embedding high-density data into a two- or three-dimensional setting for visualization. For example, if cell diversity was found to be well represented with some PCs, t-SNE will plot the cells on a two-dimensional graph in a way that preserves the relationship between cells; as a consequence, cells that are close on a multi-dimensional graph remain close together on a two-dimensional graph. UMAP is a dimension reduction technique that can be used not only for visualization but also for general nonlinear dimension reduction [67]. Sensitivity studies on these methods determined that t-SNE gave the best overall performance with the highest accuracy. On the other hand, UMAP showed the highest stability and moderate accuracy while well retaining original cohesion and separation of cell populations [68].

## 3. Recent Results on Precision Medicine Applied to Colorectal Carcinoma

Intratumoral heterogeneity is a crucial factor in tumor biology, response to therapies and patient survival [69,70]. Due to the need to characterize the phenotypes and interactions of the tumoral cell subtypes, to date molecular profiling studies have adopted a bulk approach by not identifying the signatures of distinct cell populations.

As single-cell sequencing technologies ensure a complete, unbiased analysis of cellular diversity within tumor masses, they can be used to explore the measurement somatic mutation rates, the clonal evolution of cell tumor lineages, and gain insights into chemotherapeutic drug response [71,72]. Whole genomic and transcriptomic profiling of a tumor sample shows us only average measures of cellular characteristics, thus concealing critical aspects of tumor heterogeneity.

Currently, several studies on single cells genomics and transcriptomics analysis [6,73,74] have increased existing molecular classifications of CRC by detecting new distinct subclones within a single phenotype, previously identified through standard transcriptomics [31,75].

Dai et al. generated a molecular census of tumor tissue cell types of a single CRC patient alongside with a clustering analysis to define gene expression at single-cell level. A total of 2824 cells were identified and classified into five distinct cell clusters. Each cluster was characterized by different cell markers: cluster 2 prevalently contained genes related to the major histocompatibility complex, while the remaining 4 possessed cell markers related to themselves. Gene Ontology term analysis demonstrates that cluster 1 genes were responsible for biological processes including ATP synthesis, cellular respiration, and energy derivation. Cluster 3 and 4 genes mainly supported cells by providing energy, generating extracellular matrix. Cluster 2 and 5 genes highlighted immunity functions including immune response, regulation of lymphocyte, leukocyte, and T-cell activation. Although the results of Dai et al. were obtained by a single CRC patient, they help us understand how different activated and quiescent, abnormal cellular subpopulations contribute to the initiation, maintenance, and progression of CRC disease [75]. These data could represent an interactive map of genetic interaction and might be used to identify targets to develop new therapeutic options for CRC.

Li et al. performed an scRNA-seq analysis on 11 primary CRCs and matched normal mucosa to their microenvironments. They developed a method for single-cell transcriptome analysis defined reference component analysis (RCA) based on an algorithm that improves clustering accuracy.

Seven major cell types both in normal mucosa and CRC were isolated as well as epithelial cells, fibroblasts, endothelial cells, B cells, T cells, mast cells and myeloid cells. By using RCA, nine epithelial clusters and seven epithelial cell subtypes in human normal mucosa were isolated *de novo*. These reference data allow to identify a strong enrichment of stem/TA-like cells. Two distinct types of cancer-associated fibroblasts (CAFs), and epithelial–mesenchymal transition-related genes were found to be upregulated in the tumoral CAF subpopulation. CRC defined as single type in bulk transcriptomics, might be divided into subgroups with different survival probability rates by using single-cell signatures [75].

A recent study characterized the individual cell response of CRC cell lines to genotoxic 5-fluorouracil (5FU)-induced DNA damage using a scRNA-seq approach. After 5FU treatment, the apparently single population CRC cells assume three distinctive transcriptome profiles, corresponding to diversified cell-fate responses: apoptosis, cell-cycle checkpoint, and stress resistance. Based on the group-specific expression gene patterns mediating DNA damage responses, it can be inferred how individual cells shape their transcriptome in response to DNA damage involving recurrence and chemoresistance. This might represent one of the most important challenges in current cancer treatment [76]. The identification of cell-fate-specific transcriptome patterns in in vitro experiments should promote future studies on human CRC to explore heterogeneous cancer cell responses to genotoxic chemotherapy, such as fractional killing and chemoresistant tumor recurrence.

Metastasis is a complex biological process in which tumor cells move from the primary organ site and spread to distant organs through blood circulation [77]. Various models of metastasis have been proposed: late spread, early spread, and self-seeding. In the first one, tumor cells evolve over an extended stage at the primary site and then acquire specific mutations that allow them to spread. In contrast, in the second one, cancer cells spread early, and thus primary and metastatic tumors evolve in parallel [78]. Finally, based on the self-seeding hypothesis, tumor cells spread from the primary tumor establishing distant metastatic sites and then bidirectionally return to the primary site to promote growth [79].

A general difficulty in understanding metastatic lineages depends on the large intra-tumor heterogeneity at primary and metastatic sites. Leung et al. developed a highly multiplexed single-cell DNA sequencing approach to dissect the clonal evolution during the metastatic process. They studied two CRC patients with matching liver metastases. They observed monoclonal seeding in the first patient: a single clone acquired a large number of mutations before migrating to the liver to establish the second tumor site. In the second patient, they observed polyclonal seeding: two independent clones seeded metastases to the liver after migrating from the primary tumor lineage at different time points. Single-cell data also revealed a striking independent tumor lineage that did not metastasize, and early progenitor clones with the “*first hit*” mutation in APC that subsequently gave rise to both the primary and metastatic tumors. Data from this study revealed a late-dissemination model of metastasis in both CRC patients and provided unprecedented insight into metastasis at single-cell genomic resolution [80]. Actually, despite the small number of CRC patients observed and the fact that only the liver metastatic site was examined, Leung’s study represents a preliminary confirmation that late-dissemination models of metastasis can occur in CRC but should not be contemplated as a common model for all CRC patients.

Tang et al. characterized the evolutionary pattern of metastatic CRC (mCRC) by analyzing bulk and single-cell whole-exome sequencing (scWES) data of primary and metastatic tumors from seven CRC patients. They proved that genomic profile could be better explained by using scWES than through bulk sequencing. Rare mutations highlighted by scWES were undetectable in bulk data. Several subclones have been identified in both primary and metastatic tumor cells in MSI CRC patients. Although the individual cells of each subclone share a substantial number of mutations, few subclone-specific single nucleotide variants (SNVs) could characterize different cell clones with low mutation frequencies in the entire population of tumor cells.

In MSS CRC patients, tumor cells were divided into two major cell populations from primary and metastatic lesions, that shared most SNVs and involved genes associated with CRC progression, such as TP53 and APC. Primary tumor cell populations were rich in AXIN3 and RASGRF1 genes mutation, known to be associated with tumor proliferation and invasion. In addition, 24 non-synonymous SNVs specific to metastatic cells in DNAH3, TBC1D4, CMYA5, MYO18A, PLEKHA7, and SLC19A3 genes have been identified, validating their functions in cell migration capacity [81].

Another comparison of scWES versus bulk whole-exome sequencing (bulk WES) on two CRC patients with tumor and adenomatous polyps, showed that both had monoclonal origin and shared partial mutations in the same signaling pathways; however, each showed a specific spectrum of heterogeneous somatic mutations. Adenoma and cancer further developed intratumor heterogeneity accumulating non-random somatic mutations specifically in GPCR, PI3K-Akt and FGFR signaling pathways. New driver mutations were identified that developed during the evolution of both adenoma and cancer: on one hand OR1B1 (GPCR signaling pathway) was related to adenoma evolution; on the other hand, LAMA1 (PI3K-Akt signaling pathway) and ADCY3 (FGFR signaling pathway) had a role in CRC evolution. ScWES shows causality of mutations in certain pathways that would not be detected by bulk tumor sequencing. Furthermore, it can potentially establish whether specific mutations are mutually exclusive or occur sequentially in the same subclone of cells [82].

To examine the genome, transcriptome, and methylome within CRC primary tumors and metastases, Bian et al. used a single-cell triple homology sequencing (scTrio-seq) technique [83]. The scTrio-seq technique can assess somatic copy number alterations (SCNA), as well as DNA methylation and transcriptome information simultaneously from the same single cell [84]. The authors performed a multiregional sampling and generated scTrio-seq profiles for 12 CRC patients with stage III or IV cancer. The majority of tumor cells from six of the patients analyzed were assigned to the group with abnormal activation of WNT/β-catenin and MYC signaling pathways, frequent somatic copy number alterations (SCNAs), and no hypermutation. In the 10 patients with DNA methylation data were relatively consistent within a single genetic line, single-cell SCNA profiling identified significant focal SCNAs and likely target genes. Differences in methylation profiles between primary and metastatic sites could be primarily due to differences in sub-lineage composition. No results from *de novo* methylation or demethylation during metastasis were observed. As well as providing important information about the molecular alterations that occur during CRC progression and metastasis, multicellular sequencing showed that DNA methylation levels are consistent within lineages but can differ substantially between clones [83].

To summarize, shedding light on the main mechanisms behind the development of metastasis based on the analysis of gene expression patterns at single-cell resolution should lead to tailoring individualized cancer treatment.

To this end, the study of CRC heterogeneity through identification of tumor cells subpopulations and analysis of their features by single-cell omics technologies is crucial for the comprehension of the role of these cells and might lead to identify potential new targets for clinical treatment.

Table 1 provides a summary of the most recent advances of the application of both single-cell sequencing and editing technologies into precision medicine applied to CRC patients.

Application of omics technologies on other types of cancers is opening a way to verify results also in diagnosis and treatment of other tumoral diseases, including CRC; this is the case of breast cancer (BC) thoroughly studied through single-cell omics technologies. For instance, Pinkney et al. adopted scRNA-seq to analyze the heterogeneity of lncRNA expression in vivo using Triple Negative BC (TNBC) xenografts; at the same time, they tried to assess whether lncRNA expression is sufficient to define cellular subpopulations. These authors observed that even if most lncRNAs are detectable at low levels in TNBC xenografts, a subpopulation of cells could not be defined. They showed highly heterogeneous expression patterns including global expression and subpopulation-specific expression; in addition, a hybrid pattern of lncRNAs was expressed in several but not all subpopulations [85].

LncRNAs have been progressively identified as the main group of oncology targets acting as drivers in cancer, and are also being studied as clinical biomarkers [86]. LncRNAs link with biological molecules, as well as with DNA, mRNAs, miRNAs and proteins, modulating epigenetic, transcriptional, post-transcriptional, translational and post-translational events in gene expression [87,88]. LncRNAs have been observed to be of interest in cancer, but little is known about their expression in cell subpopulations. Further investigation may determine whether expression of specific lncRNAs contribute to specific cell populations features; they might have a role also in invasion and/or proliferation, considering that lncRNAs have been described as drivers of these processes [89]. Therefore, the spatial distribution of lncRNAs within a patient’s cancer tissues might identify the potential of subclone-specific lncRNAs as new therapeutic targets and/or biomarkers.

Zhang et al. performed a single-cell RNA- and ATAC-sequencing to examine the immune cell dynamics in advanced TNBC patients treated with paclitaxel or paclitaxel plus atezolizumab (anti-PD-L1). High levels of baseline CXCL13+ T cells linked to macrophage proinflammatory features might predict responses to a drug combination. In patients responsive to drug combination, an increase of lymphoid tissue inducer cells, follicular B cells, CXCL13+ T cells, and type 1 dendritic cells was detected. The latter decreased after paclitaxel monotherapy [90]. These data suggest the role of CXCL13+ T cells in the responses to anti-PD-L1 therapies.

Immune checkpoint blockade (ICB) targeting PD-1/PD-L1 signaling axis and its use has achieved significant responses in cancer patients, although the mechanisms underlying ICB resistance have not been fully understood [91,92]. Thus, the advances in single-cell technologies enable to characterize the basic properties of tumor-infiltrating immune cells to determine their role in immune responses, antitumor immunity, and immunotherapies.

**Table 1 medicina-57-01257-t001:** Summary of advances of single-cell sequencing and editing technologies into precision medicine in the colorectal cancer.

Sample Type	Technology	Findings	Implications	Ref.
1 patients 2824 sc	scRNA-seq	-5 distinct cell subsets were identified consisting of: immune cells, related to the major histocompatibility complex genes, related to genes serving to stabilize the cell, energy transportation and cell regulation, TSPAN6, PFDN4, and TIMM13, majored in breakdown of extracellular matrix and tissues remodeling, and genes involved in cancer, WFDC2 -cluster 1 and 3 revealed biological processes genes, including ATP synthesis, cellular respiration, oxidative phosphorylation, and mitochondrion organization -cluster 2 and 5 revealed biological process genes, consisting of activation, positive regulation, response to stress, cellular response, and cell adhesion -cluster 4 revealed biological processes responsible for extracellular matrix organization, response to stress, locomotion, cell migration, and cell motility	-provides insight into the heterogeneity of CRC and which genes within each cluster serve different functions	[93]
11 patients 7 cell lines CRC 590 patient-derived sc 561 cell line-derived sc	scRNA-seq, reference component analysis algorithm	-scRNA-seq generated further sub-classification of CRC subtypes found by bulk RNA-seq with prognostic significance based on their single-cell signatures	-scRNA-seq could enable clinically relevant patient stratification	[75]
3 cell lines CRC	scRNA-seq	-transcriptomic characterization of CRC cell lines response to 5-fluorouracil (5FU)-induced DNA damage -three distinct transcriptome phenotypes were assumed by CRC cells, with different cell-fate responses: apoptosis, cell-cycle checkpoint, and stress resistance	-understanding of the heterogeneous DNA damage responses involved in fractional killing and chemoresistance	[76]
2 patients 360 sc and bulk primary tumor and liver metastasis	scNGS, bulk WES	-the single-cell and bulk analyses were highly concordant -monoclonal and polyclonal seeding were found -rare cell subpopulations were associated with progression and metastasis -a late-dissemination model was highly concordant between primary tumor and liver metastasis samples	-the late-dissemination model suggests that early surgical intervention could prevent metastasis	[80]
7 patients 321 sc and bulk primary tumor and liver metastasis	scWES, bulk WES	-low genomic divergence between paired primary and metastatic cancers were found in bulk data -scWES data defined two separate cell populations, indicative of the diverse evolutionary trajectories between primary and metastatic tumor cells. -rare mutations were identified using single-cell technology that were overlooked in bulk data	-validation of functions of different metastatic subclone-specific-mutated genes in cell migration	[81]
2 patients 96 sc (adenomatous polyp and CRC)	scWES, bulk WES	-adenoma and cancer have monoclonal origin with subsequent subclonal evolution -adenoma and cancer showed a specific spectrum of heterogeneous somatic mutations -novel driver mutations that developed during adenoma and cancer evolution, in OR1B1 (GPCR signaling pathway) for adenoma evolution; LAMA1 (PI3K-Akt signaling pathway) and ADCY3 (FGFR signaling pathway) for CRC evolution	-scWES provides evidence for the importance of mutations in certain pathways that would not be so apparent from bulk sequencing of tumors	[82]
12 patients 1900 sc and bulk multi-regional	scTrio-seq, bulk multi-regional WGS	-cancer cells were classified into several genetic subclones -primary tumor showed higher subclonality than metastatic tumour -DNA methylation profiles were stable within a single genetic lineage	-single-cell multiomics sequencing can trace epigenomic and transcriptomic dynamics during progression and metastasis	[83]
2 patients CRC clonal tumor organoids	3D Live-Seq (a protocol that integrates live-cell imaging of tumor organoid outgrowth and WGS of each imaged cell to reconstruct evolving tumor cell karyotypes across consecutive cell generations)	-reveals the genomic consequences of CIN across consecutive cell generations -single-cell sequencing data displayed several de novo CNAs across three lineages -mis-segregation of chromosome 7 displays the highlighted branch within the mitotic tree	-mapping the temporal dynamics and patterns of karyotype diversification in cancer enables reconstructions of evolutionary paths to malignant fitness	[94]
Cell lines CRC tumor, stroma, adjacent normal, lung metastasis	quantitative micro-engraving	-single cells exhibit a range of secretory phenotypes for CXCL1, CXCL5, and CXCL8 -secretions of ELR+ CXC chemokines were found from thousands of single CRC and stromal cells -CRC and stromal cells exhibit polyfunctional heterogeneity in the combinations and magnitudes of secretions for these chemokines -discordances exist between secretory states measured and gene expression for these chemokines among single cells	-these measures suggest that secretory states among tumor cells are complex and can dynamically evolve -heterogeneous release of these chemokines by individual cells promotes a robust signaling network within the tumor microenvironment	[95]
14 patients 336 cells each phenotypic population	scPCR gene-expression analysis	-CRC tissues contain distinct cell populations whose transcriptional identities mirror those of the different cellular lineages in healthy colon -perturbations in gene expression programs linked to multi-lineage differentiation strongly associate with patient survival -development of two-gene classifier systems (KRT20 vs. CA1, MS4A12, CD177, SLC26A3) that predict clinical outcomes with hazard-ratios superior to pathological grade	-development of a simple and quantitative nature two-gene scoring system	[96]
2 patients 88 sc rectal cancer	WES, scWGS multi-region	-genomic heterogeneity was observed between the two patients, and the degree of ITH increased when analyzed at single-cell level -SCNAs were early events in cancer development -single-cell sequencing revealed mutations and SCNAs which were hidden in bulk sequencing	-each tumor possesses its own architecture, which may result in different diagnosis, -prognosis, and drug responses	[97]
2 patients 47 sc cancer stem and differentiated tumor	scWGS	-CD45− EpCAM^high^ CD44+ CSCs and CD45− EpCAM^high^ CD44− differentiated tumor cells had similar SCNA profiles -the similarity of ubiquitous SCNAs between the CSCs and DTCs might have arisen from lineage differentiation	-the possibility of a monoclonal CSC phenotype is supported	[98]
3 patients organoid from multiple sc CRC and normal mucosa	scWGS	-significant intra-tumor clonal heterogeneity with specific mutational signatures were identified organoids treated with chemotherapeutic and targeted agents, even derived from the same patient, exhibited differential responses independent of their mutational signatures	-substantial increases in somatic mutation rate compared to normal colorectal cells -genetic diversification of each cancer is accompanied by pervasive, stable, and inherited differences in biological states of individual cancer cells	[6]

scRNA-seq: single-cell RNA-seq, sc: single-cell, CRC: colorectal carcinoma, scNGS: single-cell next generation sequencing, scWES: single-cell whole-exome sequencing, WGS: whole genome sequencing, SCNAs: somatic copy number alterations, TME: tumor microenvironment, S-TAM: small tumor-associated macrophages, L-TAM: large tumor-associated macrophages, CCDGs: cell cluster deregulated genes, CIN: chromosomal instability, CAN: copy-number alterations, ITH: intratumor heterogeneity.

## 4. Conclusions

### 4.1. Future Perspectives in Methodologies

In-depth knowledge of the cells of interest is crucial to properly manage genomic data and make decisions of clinical impact based on standardized measurements and accurate and reproducible quality controls. The use of scRNA-seq provides one of the most innovative methods for addressing biological and medical questions concerning the underlying processes of various developmental, physiological, and disease systems. However, new programs and implementations of scRNA-seq methodologies have been started in recent years but further advances in both technology and specific approaches to use them are certainly warranted.

The deployment of a number of processes will make it possible to extend the analysis of scRNA-seq studies not only on fresh material, but also on cryopreserved and fixed tissue samples aiming at introducing this technique into the clinical practice. Volume reduction and diffusion of techniques based primarily on microfluidics platforms should reduce costs at the same time leading to a standardized and simplified use of different devices.

However, one of the current challenges is the creation of standardized collections and data catalogues from single cells due to the fact that the number of samples used so far in studies is small. Such analysis, in fact, requires a minimum/sufficient number of cells to ensure that all cell types are represented. Only a bioinformatician with experience in single-cell sequencing will be able to generate analyses that can be used to make meaningful biological inferences by choosing appropriate cutoffs for applied algorithms and avoiding misleading results. Currently, there are limited standardization protocols and guidelines on standards (i.e., quality control, removal of technical artifacts, etc.).

Furthermore, development of single-cell gene expression maps for all tissues will be necessary, as it occurred in bulk transcriptomics evolution. Many studies, in fact, will benefit from these easily accessible archives that reduce the costs of comparison and replication in normal tissues; at the same time, significant advances in bioinformatics and computational methods thanks to data sharing are expected.

Thus, the new challenge will be represented by the use of a true inter-omic and multidisciplinary approach that will lead to a comprehensive examination of individual cells; this will be achieved by characterizing the genome, epigenome, proteome and metabolome while simultaneously examining the tumor microenvironment, its immunological characteristics and the impact of pharmacogenomics; in addition, a clear picture of tumor development will be given, together with cancer evolution and interactions. It will be crucial to address genetic changes in the early stages of tumorigenesis deployment and how transcriptional subpopulations evolve into malignancy in later stages of tumor progression.

The robustness of NGS systems in exploring heterogeneity, at genome and transcriptome scale, will validate ITH variability and might determine the discovery of novel targeted drugs; predictive biomarkers for individualized drug-oriented therapies might also be developed. Pharmacogenomic profiling might predict response to chemotherapy by correlating it with immune cell regulatory values that affect CRC survival mechanisms. Future CRC studies employing comparison of primary, metastatic tumor ITH and liquid biopsies might offer elucidating suggestions on the origins and evolution of genomic subclones responsible for drug resistance and recurrence.

### 4.2. Clinical Implications of Intra-Tumoral Heterogeneity in CRC

CRC is extensively marked by phenomena of inter- and intra-tumor heterogeneity, spatial and temporal differences regarding phenotypic and genotypic aspects, influencing recurrence and therapeutic response and having a strong poor impact on CRC patient’s outcome.

Until now, genomic and transcriptome analyses on bulk tumor cell populations have helped to explain tumor heterogeneity and also allowed to classify them into subgroups with distinct molecular, morphological, and clinical features [99]. The application of techniques capable of examining molecular aberrations at the single-cell level within a complex tumor population should refine the existing CRC classification system. In addition, scRNA-seq could identify predictive markers for CRC prognosis.

Metastatic progression is linked to the majority of CRC-related deaths [100]. In patients at stage I, the five-year survival rate is 90%, but a drastic reduction of slightly more than 10% is observed when cancer patients reach stage IV [101]. Approximately 20% of CRC patients already have metastases at diagnosis, and they are generally incurable [100]. Although anti-EGFR therapies are available for RAS wild-type CRC patients, and anti-VEGF, anti-VEGFR, recombinant fusion protein and multi-kinase inhibitor were applied in CRC patients with RAS mutation [102], unresponsiveness was seen in CRC patients with BRAF and PIK3CA mutations [103]. Undoubtedly, drug development and techniques to be used in identifying the complex heterogeneity of mCRC represent an unmet clinical need. Single-cell omics represent an important tool to identify therapeutic targets for personalized cancer medicine compared with bulk transcriptomics. In addition, single-cell resolution molecular aberrations could shed light on the mechanisms underlying metastasis development [104,105,106]. Finally, the ability to estimate presence of rare malignant chemical-resistant carcinoma cells in removed tumors will be increased to guide treatment decisions; at the same time exploration of immune cell responses and environmental influences will provide molecular data to give support during the diagnostic process as well as in disease progression, and treatment course.

## Figures and Tables

**Figure 1 medicina-57-01257-f001:**
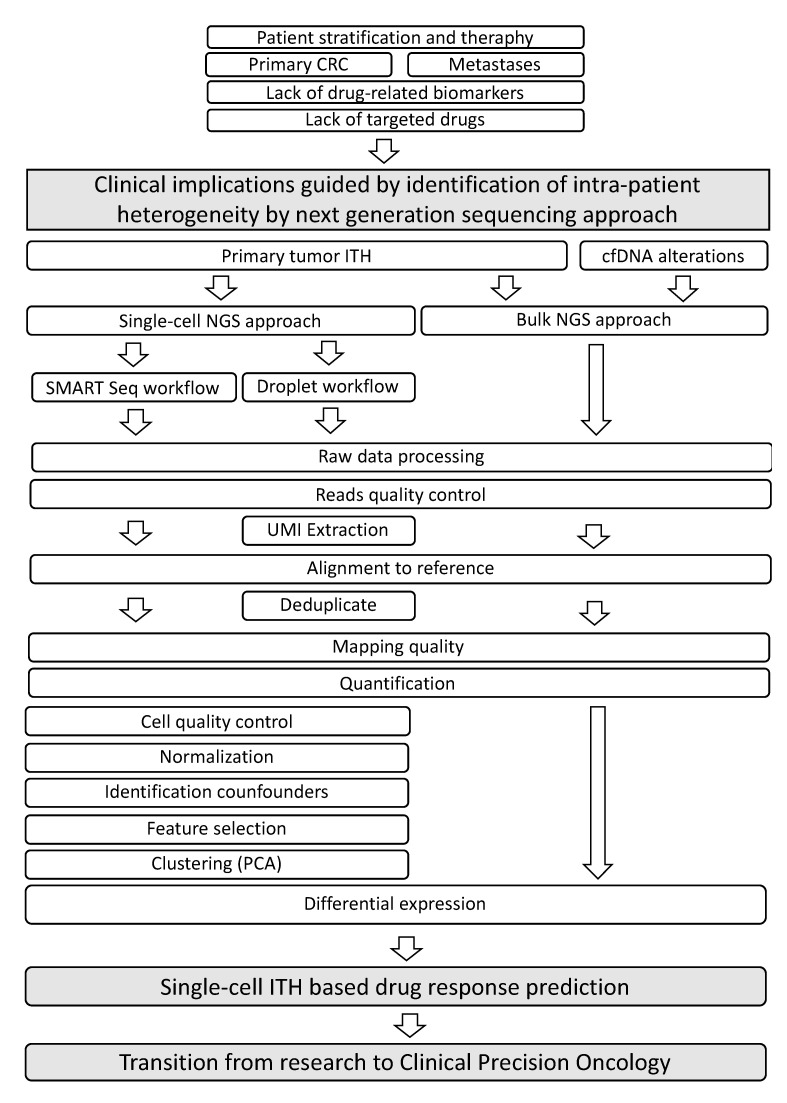
Brief outline of the state of the art of colorectal cancer management, issues to be addressed and potential solutions proposed by recent technologies for exploring genome and transcriptome alterations by mass and single-cell sequencing.

## Data Availability

Not applicable.

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
