# Peer review of "A Portrait of Intratumoral Genomic and Transcriptomic Heterogeneity at Single-Cell Level in Colorectal Cancer"

_medicina, 2021, doi:10.3390/medicina57111257_

Round 1

Reviewer 1 Report

The authors try to overview (with a narrative review) the current knowledge concerning single-cell analysis in colorectal cancer. The general idea is interesting, however, the article has several major limitations that impact negatively of its overall scientific quality

1-The text is very hard to follow and it is written in very poor English. The whole text is built with very long sentences and full of unnecessary adjectives that overemphasize the theoretical importance of any given subject. Nevertheless, regardless of the section, most of the body of the text is made of a long list of examples that are never considered as a whole (reviewed) or used to show a specific concept.

2-The sections are full of medical cliches that are only rarely seriously and scientifically examined/discussed (e.g. it is often cited a general clinical need of deeper genomic analyses, without a description of its real clinical impact (e.g. studies on differences in survivals or types/rates of therapy response).

3-Many sections include paragraphs that are completely out of topic (e.g. the very long section 2. lines 110-217 dealing with a sort of methodological review or lines 187-193 on myocardial mitochondria).

I also find out of topic the long section on the tumor microenvironment. It is a different topic with different complexity. Moreover, its description cannot be addressed citing randomly some T cell and myeloid subtypes without considering their functional specificity. The results listed in that section are again mentioned in a list without further integration/review.

4-Finally, but importantly, this narrative review lacks a section describing how the authors search, selected, and analyzed their data, as suggested in the international guidelines to medical reviews.

The authors should try to rewrite the article including:

  1. a brief introduction describing the state of the art, utility, and clinical impact of single-cell genomic and transcriptomic analyses used to study tumor heterogeneity.
  2. a material and methods section explaining how they select the analyzed articles.
  3. a section describing the current knowledge on single-cell genomics/transcriptomics in CRC heterogeneity (avoid including description of microenvironment)
  4. a discussion of the biological and clinical (incl prognostic & predictive) significance of these data.

Author Response

  1. The text is very hard to follow and it is written in very poor English. The whole text is built with very long sentences and full of unnecessary adjectives that overemphasize the theoretical importance of any given subject. Nevertheless, regardless of the section, most of the body of the text is made of a long list of examples that are never considered as a whole (reviewed) or used to show a specific concept.

According to the reviewer's comment, a complete critical revision of the entire manuscript was done trying to finalize the discussion and make it more concise. It was not feasible to consider the results of the different studies as a whole, because they are very heterogeneous. For example, the generation of data ranges from RNAseq to whole genome and whole exome sequencing and consequently the data are not directly comparable and reassemble. The approach used is extremely different: some of them for example use only a small number of tissue samples with an approach aimed at a specific purpose. The manuscript has been completely proofread by an experienced native speaker writer.

  1. The sections are full of medical cliches that are only rarely seriously and scientifically examined/discussed (e.g. it is often cited a general clinical need of deeper genomic analyses, without a description of its real clinical impact (e.g. studies on differences in survivals or types/rates of therapy response).

Thanks for the reviewer’s comment. Now, the single-cell omics data available in the literature are experimental results coming from basic research. Despite the last few years have seen significant technological progress in single cell technologies, they are not yet standardized and usable on a large scale to have sufficient statistical power, to be introduced in clinical trials. A real clinical impact on survival and response on therapies cannot be assessed. We could hypothesize the impact on the cancer progression characterized by recurrence and resistance to therapeutic options for CRC. For this reason, we include in the conclusion a subsection on clinical implications of intra-tumoral heterogeneity in CRC (see pages 12-13  lanes 532-559).

  1. Many sections include paragraphs that are completely out of topic (e.g. the very long section 2. lines 110-217 dealing with a sort of methodological review or lines 187-193 on myocardial mitochondria).

Thanks for the reviewer’s comment. Considering the importance of new omics technologies, the lack of standardization of omics protocols and the emerging of new approaches and algorithms, especially in the downstream steps of scRNAseq analysis (normalization, clustering and dimensionality), we assumed that to include a section which summarizes and describes the methodologies that are mainly applied would be useful to members of the scientific community who are approaching to these issues. We delete the sentence regarding myocardial mitochondria.

  1. I also find out of topic the long section on the tumor microenvironment. It is a different topic with different complexity. Moreover, its description cannot be addressed citing randomly some T cell and myeloid subtypes without considering their functional specificity. The results listed in that section are again mentioned in a list without further integration/review.

Thanks for the reviewer’s comment. We took into consideration the microenvironment and the composition of the immune cells given their importance in inducing the tumor heterogeneity. The characteristics of solid tumors are not only the result of clonal growth of cells with genetic mutations, but also of epigenetic alterations induced by physical and chemical signals from the tumor microenvironment.

  1. Finally, but importantly, this narrative review lacks a section describing how the authors search, selected, and analyzed their data, as suggested in the international guidelines to medical reviews.

The authors should try to rewrite the article including:

a brief introduction describing the state of the art, utility, and clinical impact of single-cell genomic and transcriptomic analyses used to study tumor heterogeneity.

a material and methods section explaining how they select the analyzed articles.

a section describing the current knowledge on single-cell genomics/transcriptomics in CRC heterogeneity (avoid including description of microenvironment)

a discussion of the biological and clinical (incl prognostic & predictive) significance of these data.

Thank you for the reviewer's comment. We consider our review not a medical review but a descriptive review that includes experimental results available in the omics research literature. We aimed to give an overview of the state of the art in the evolution of techniques and results in the application of single cells in colorectal cancer.

Reviewer 2 Report

In this review, De Miglio and colleagues discuss on intratumoral and microenvironment heterogeneity in colorectal cancer (CRC) from a single-cell analysis perspective. The manuscript is interesting and provides state-of-the-art information about single-cell genomics and transcriptomics in this type of malignancy.

Minor suggestions for improving the already good quality of the review are the following:

  1. Authors provided a nice introduction about the methodology and analysis of single cell data. I suggest to include together with PCA and t-SNE, some information about UMAP as a graphical way to reduce dimensionality of single-cell data.

  1. Authors could add some critical comments or limitations to some of the works reviewed in the manuscript.

  1. As a related comment to the above, authors could conclude with some single-cell cutting-edge analyses performed in other cancer types that could be performed in CRC and could add more knowledge that still is missing on this particular field.

  1. Please check if reference number 73 is correct.

  1. In line 358, authors referred to “Zhou et al” work. However, this article is not linked to any reference and in fact, the article is missing in the reference list. In the table is probably indicated as reference number 110 but again, this reference is missing.

Author Response

-Authors provided a nice introduction about the methodology and analysis of single cell data. I suggest to include together with PCA and t-SNE, some information about UMAP as a graphical way to reduce dimensionality of single-cell data.

We are grateful for the comment, and we agree with the reviewer’s suggestion. We added information about UMAP in the methodologies section (see pag. 5 lanes 194-198 to pages 5-6 lanes 203-208)

-Authors could add some critical comments or limitations to some of the works reviewed in the manuscript.

Accordingly, with the reviewer’s comments, we added comments and/or limitations of the works reviewed in the manuscript (see pag. 6 lanes 234-239; see pag. 7 lanes 260-263; see pag. 7 lanes 284-28, see pag. 8 lanes 333-335; see pag. 9 lanes 359-363-239; see pages 10-11  lanes 437-442).

-As a related comment to the above, authors could conclude with some single-cell cutting-edge analyses performed in other cancer types that could be performed in CRC and could add more knowledge that still is missing on this particular field.

Accordingly, with the reviewer’s comments, we added some interesting results on single-cell omics analysis related to breast cancer that could be performed in CRC giving attractive data in the field of epigenomic and immunotherapy (see pag. 11  lanes 445-478)

-Please check if reference number 73 is correct.

We are grateful for the comment. We eliminated the reference 73, it was wrong.

-In line 358, authors referred to “Zhou et al” work. However, this article is not linked to any reference and in fact, the article is missing in the reference list. In the table is probably indicated as reference number 110 but again, this reference is missing.

            We are grateful for the comment. We included the reference related to Zhou et al, in the text and table.

Round 2

Reviewer 1 Report

    1.  The authors add around 1000 words to an already very long paper. It is too long and still hard to follow. 
    2. Medicine reviews may be narrative or systematic. I understand the purpose of the authors to write a narrative review. However, it is mandatory to add a "material and methods" section to qualify this article as a review (otherwise it may be e.g. a selection of works that the authors may want to highlight over a much wider literature). Therefore there is a need for reproducibility that is unmet. This is of special importance, given that even the authors state that the literature on the subject is incredibly heterogeneous (for this reason, it would be better to write a systematic rather than narrative review).
    3. The authors edited the English language. However, they do not address the very problem (long sentences).
    4. I understand the author's difficulty to take as a whole many different preclinical studies. However, I have some concerns regarding the utility of an article made by a list of heterogeneous studies. I believe that the authors should try to focus on a scientific/clinical question to answer with this review. The actual structure of the article in my opinion is not suitable for publication in a medical journal.
    5. Still, many sections include paragraphs that are out of topic (e.g. a long section added on pg 11 lines from 445). Can the authors explain the significance of such a long paragraph on breast cancer in an article on CRC? (If members of the scientific community want to gain general information regarding omics they will refer to the specific general literature on the topic). 
    6. Regarding the tumor microenvironment (which I study) I understand its importance, but is still not the topic of your article entitled "[...]intratumoral genomic and transcriptomic heterogeneity [...] Moreover technically speaking the paragraph on tumor immune microenvironment only focuses on limited and selected (How?) aspects of the role of SOME immune cells in cancers. Please, just delete it. 
    7. Finally, I will highlight again the excessive length of the review, and the large amount of space dedicated to aspects of oncology that are out of topic. 

    In the present form (very similar to the previously submitted one) the article is not scientifically sounding, not reproducible and In my opinion of very low clinical and biological impact in medical research.

    Please refer to both the previous and the current reviewer's comments to modify your submission.

Author Response

  1. The authors add around 1000 words to an already very long paper. It is too long and still hard to follow.

We can understand the reviewer's point of view, but we had to add additional sentences in response to a specific request of reviewer 2 asking:

” As a related comment to the above, authors could conclude with some single-cell cutting-edge analyses performed in other cancer types that could be performed in CRC and could add more knowledge that still is missing on this particular field.

Our Response: Accordingly, with the reviewer’s comments, we added some interesting results on single-cell omics analysis related to breast cancer that could be performed in CRC giving attractive data in the field of epigenomic and immunotherapy …..”.

  1. Medicine reviews may be narrative or systematic. I understand the purpose of the authors to write a narrative review. However, it is mandatory to add a "material and methods" section to qualify this article as a review (otherwise it may be e.g. a selection of works that the authors may want to highlight over a much wider literature). Therefore there is a need for reproducibility that is unmet. This is of special importance, given that even the authors state that the literature on the subject is incredibly heterogeneous (for this reason, it would be better to write a systematic rather than narrative review).

Thanks for the reviewer's comment. We looked at the number of reviews published in "Medicina" in the last issue (Volume 57, Issue 10 (October 2021) – 139 articles): out of 34 reviews on only less than 50% (14 out of 34) included the Material and Methods section and they are all systematic reviews. Our intent was to write a narrative review taking into consideration both technical and genomic/clinical aspects. Considering the very few articles published on CRC single-cell omics technologies, which are the subject of the review, we do not consider it necessary.

  1. The authors edited the English language. However, they do not address the very problem (long sentences).

Thanks for the reviewer’s comment. We have made a second complete revision of the manuscript proofread by an experienced native speaker writer, which has revised the long sentences.

  1. I understand the author's difficulty to take as a whole many different preclinical studies. However, I have some concerns regarding the utility of an article made by a list of heterogeneous studies. I believe that the authors should try to focus on a scientific/clinical question to answer with this review. The actual structure of the article in my opinion is not suitable for publication in a medical journal.

Thank you for the reviewer's comment. Other reviews published in Medicina, especially those involving omics data also have a similar structure to the one we have adopted. Heterogeneity, as stated in the previous response, comes from experimental results from basic research that are aimed at specific purposes and consequently adopt specific approaches. Despite this limitation, it is our opinion that our article can be useful to the scientific community because it tries, in a summarized way, to give a global picture of the state of the art. Reviewer 2, for example, considers positively our work and useful for the scientific community.

  1. Still, many sections include paragraphs that are out of topic (e.g. a long section added on pg 11 lines from 445). Can the authors explain the significance of such a long paragraph on breast cancer in an article on CRC? (If members of the scientific community want to gain general information regarding omics they will refer to the specific general literature on the topic).

Thank you for the reviewer's comment. We have added a paragraph on breast cancer at the specific request of reviewer 2 (see details in answer n.1)

  1. Regarding the tumor microenvironment (which I study) I understand its importance but is still not the topic of your article entitled "[...]intratumoral genomic and transcriptomic heterogeneity [...] Moreover technically speaking the paragraph on tumor immune microenvironment only focuses on limited and selected (How?) aspects of the role of SOME immune cells in cancers. Please, just delete it.

We have deleted the subsection “Heterogeneity of stromal and immune cells in the tumor microenvironment identified by single cell omics technologies”. Although, we have kept this subsection because in the first revision the Review 2 also appreciated this part of the work. His comment was “In this review, De Miglio and colleagues discuss on intratumoral and microenvironment heterogeneity in colorectal cancer (CRC) from a single-cell analysis perspective. The manuscript is interesting and provides state-of-the-art information about single-cell genomics and transcriptomics in this type of malignancy.”

  1. Finally, I will highlight again the excessive length of the review, and the large amount of space dedicated to aspects of oncology that are out of topic.

We have tried to reduce substantially, from a linguistic and scientific aspect, our review by removing what was explicitly asked for, but we believe that some aspects are needed to give a comprehensive discussion of the review topic.